# Riemannian approach to batch normalization

**Minhyung Cho**     **Jaehyung Lee**
Applied Research Korea, Gracenote Inc.
mhyung.cho@gmail.com     jaehyung.lee@kaist.ac.kr

## Abstract

Batch Normalization (BN) has proven to be an effective algorithm for deep neural network training by normalizing the input to each neuron and reducing the internal covariate shift. The space of weight vectors in the BN layer can be naturally interpreted as a Riemannian manifold, which is invariant to linear scaling of weights. Following the intrinsic geometry of this manifold provides a new learning rule that is more efficient and easier to analyze. We also propose intuitive and effective gradient clipping and regularization methods for the proposed algorithm by utilizing the geometry of the manifold. The resulting algorithm consistently outperforms the original BN on various types of network architectures and datasets.

## 1 Introduction

Batch Normalization (BN) [1] has become an essential component for breaking performance records in image recognition tasks [2, 3]. It speeds up training deep neural networks by normalizing the distribution of the input to each neuron in the network by the mean and standard deviation of the input computed over a mini-batch of training data, potentially reducing *internal covariate shift* [1], the change in the distributions of internal nodes of a deep network during the training.

The authors of BN demonstrated that applying BN to a layer makes its forward pass invariant to linear scaling of its weight parameters [1]. They argued that this property prevents model explosion with higher learning rates by making the gradient propagation invariant to linear scaling. Moreover, the gradient becomes inversely proportional to the scale factor of each weight parameter. While this property could stabilize the parameter growth by reducing the gradients for larger weights, it could also have an adverse effect in terms of optimization since there can be an infinite number of networks, with the same forward pass but different scaling, which may converge to different local optima owing to different gradients. In practice, networks may become sensitive to the parameters of regularization methods such as weight decay.

This ambiguity in the optimization process can be removed by interpreting the space of weight vectors as a Riemannian manifold on which all the scaled versions of a weight vector correspond to a single point on the manifold. A properly selected metric tensor makes it possible to perform a gradient descent on this manifold [4, 5], following the gradient direction while staying on the manifold. This approach fundamentally removes the aforementioned ambiguity while keeping the invariance property intact, thus ensuring stable weight updates.

In this paper, we first focus on selecting a proper manifold along with the corresponding Riemannian metric for the scale invariant weight vectors used in BN (and potentially in other normalization techniques [6, 7, 8]). Mapping scale invariant weight vectors to two well-known matrix manifolds yields the same metric tensor, leading to a natural choice of the manifold and metric. Then, we derive the necessary operators to perform a gradient descent on this manifold, which can be understood as a constrained optimization on the unit sphere. Next, we present two optimization algorithms - corresponding to the Stochastic Gradient Descent (SGD) with momentum and Adam [9] algorithms. An intuitive gradient clipping method is also proposed utilizing the geometry of this space. Finally,

we illustrate the application of these algorithms to networks with BN layers, together with an effective regularization method based on variational inference on the manifold. Experiments show that the resulting algorithm consistently outperforms the original BN algorithm on various types of network architectures and datasets.

## 2 Background

### 2.1 Batch normalization

We briefly revisit the BN transform and its properties. While it can be applied to any single activation in the network, in practice it is usually inserted right before the nonlinearity, taking the pre-activation $z = \boldsymbol{w}^\top \boldsymbol{x}$ as its input. In this case, the BN transform is written as

$$\text{BN}(z) = \frac{z - \text{E}[z]}{\sqrt{\text{Var}[z]}} = \frac{\boldsymbol{w}^\top (\boldsymbol{x} - \text{E}[\boldsymbol{x}])}{\sqrt{\boldsymbol{w}^\top \boldsymbol{R}_{\boldsymbol{xx}} \boldsymbol{w}}} = \frac{\boldsymbol{u}^\top (\boldsymbol{x} - \text{E}[\boldsymbol{x}])}{\sqrt{\boldsymbol{u}^\top \boldsymbol{R}_{\boldsymbol{xx}} \boldsymbol{u}}} \tag{1}$$

where $\boldsymbol{w}$ is a weight vector, $\boldsymbol{x}$ is a vector of activations in the previous layer, $\boldsymbol{u} = \boldsymbol{w}/|\boldsymbol{w}|$, and $\boldsymbol{R}_{\boldsymbol{xx}}$ is the covariance matrix of $\boldsymbol{x}$. Note that $\text{BN}(\boldsymbol{w}^\top \boldsymbol{x}) = \text{BN}(\boldsymbol{u}^\top \boldsymbol{x})$. It was shown in [1] that

$$\frac{\partial \text{BN}(\boldsymbol{w}^\top \boldsymbol{x})}{\partial \boldsymbol{x}} = \frac{\partial \text{BN}(\boldsymbol{u}^\top \boldsymbol{x})}{\partial \boldsymbol{x}} \quad \text{and} \quad \frac{\partial \text{BN}(z)}{\partial \boldsymbol{w}} = \frac{1}{|\boldsymbol{w}|} \frac{\partial \text{BN}(z)}{\partial \boldsymbol{u}} \tag{2}$$

illustrating the properties discussed in Sec. 1.

### 2.2 Optimization on Riemannian manifold

Recent studies have shown that various constrained optimization problems in Euclidian space can be expressed as unconstrained optimization problems on submanifolds embedded in Euclidian space [5]. For applications to neural networks, we are interested in Stiefel and Grassmann manifolds [4, 10]. We briefly review them here. The Stiefel manifold $\mathcal{V}(p, n)$ is the set of $p$ ordered orthonormal vectors in $\mathbb{R}^n (p \leq n)$. A point on the manifold is represented by an $n$-by-$p$ orthonormal matrix $Y$, where $Y^\top Y = I_p$. The Grassmann manifold $\mathcal{G}(p, n)$ is the set of $p$-dimensional subspaces of $\mathbb{R}^n (p \leq n)$. It follows that $\text{span}(A)$, where $A \in \mathbb{R}^{n \times p}$, is understood to be a point on the Grassmann manifold $\mathcal{G}(p, n)$ (note that two matrices $A$ and $B$ are equivalent if and only if $\text{span}(A) = \text{span}(B)$). A point on this manifold can be specified by an arbitrary $n$-by-$p$ matrix, but for computational efficiency, an orthonormal matrix is commonly chosen to represent a point. Note that the representation is not unique [5].

To perform gradient descent on those manifolds, it is essential to equip them with a Riemannian metric tensor and derive geometric concepts such as geodesics, exponential map, and parallel translation. Given a tangent vector $v \in T_x \mathcal{M}$ on a Riemannian manifold $\mathcal{M}$ with its tangent space $T_x \mathcal{M}$ at a point $x$, let us denote $\gamma_v(t)$ as a unique geodesic on $\mathcal{M}$, with initial velocity $v$. The exponential map is defined as $\exp_x(v) = \gamma_v(1)$, which maps $v$ to the point that is reached in a unit time along the geodesic starting at $x$. The parallel translation of a tangent vector on a Riemannian manifold can be obtained by transporting the vector along the geodesic by an infinitesimally small amount, and removing the vertical component of the tangent space [11]. In this way, the transported vector stays in the tangent space of the manifold at a new point.

Using the concepts above, a gradient descent algorithm for an abstract Riemannian manifold is given in Algorithm 1 for reference. This reduces to the familiar gradient descent algorithm when $\mathcal{M} = \mathbb{R}^n$, since $\exp_{y_{t-1}}(-\eta \cdot h)$ is given as $y_{t-1} - \eta \cdot \nabla f(y_{t-1})$ in $\mathbb{R}^n$.

---

**Algorithm 1** Gradient descent of a function $f$ on an abstract Riemannian manifold $\mathcal{M}$

---

**Require:** Stepsize $\eta$
Initialize $y_0 \in \mathcal{M}$
for $t = 1, \cdots, T$
    $h \leftarrow \text{grad} f(y_{t-1}) \in T_{y_{t-1}} \mathcal{M}$      where $\text{grad} f(y)$ is the gradient of $f$ at $y \in \mathcal{M}$
    $y_t \leftarrow \exp_{y_{t-1}}(-\eta \cdot h)$

---

# 3   Geometry of scale invariant vectors

As discussed in Sec. 2.1, inserting the BN transform makes the weight vectors $\boldsymbol{w}$, used to calculate the pre-activation $\boldsymbol{w}^\top \boldsymbol{x}$, invariant to linear scaling. Assuming that there are no additional constraints on the weight vectors, we can focus on the manifolds on which the scaled versions of a vector collapse to a point. A natural choice for this would be the Grassmann manifold since the space of the scaled versions of a vector is essentially a one-dimensional subspace of $\mathbb{R}^n$. On the other hand, the Stiefel manifold can also represent the same space if we set $p = 1$, in which case $\mathcal{V}(1, n)$ reduces to the unit sphere. We can map each of the weight vectors $\boldsymbol{w}$ to its normalized version, i.e., $\boldsymbol{w}/|\boldsymbol{w}|$, on $\mathcal{V}(1, n)$. We show that popular choices of metrics on those manifolds lead to the same geometry.

Tangent vectors to the Stiefel manifold $\mathcal{V}(p, n)$ at $Z$ are all the $n$-by-$p$ matrices $\Delta$ such that $Z^\top \Delta + \Delta^\top Z = 0$ [4]. The canonical metric on the Stiefel manifold is derived based on the geometry of quotient spaces of the orthogonal group [4] and is given by

$$g_s(\Delta_1, \Delta_2) = \text{tr}(\Delta_1^\top (I - ZZ^\top/2)\Delta_2) \tag{3}$$

where $\Delta_1, \Delta_2$ are tangent vectors to $\mathcal{V}(p, n)$ at $Z$. If $p = 1$, the condition $Z^\top \Delta + \Delta^\top Z = 0$ is reduced to $Z^\top \Delta = 0$, leading to $g_s(\Delta_1, \Delta_2) = \text{tr}(\Delta_1^\top \Delta_2)$.

Now, let an $n$-by-$p$ matrix $Y$ be a representation of a point on the Grassmann manifold $\mathcal{G}(p, n)$. Tangent vectors to the manifold at $\text{span}(Y)$ with the representation $Y$ are all the $n$-by-$p$ matrices $\Delta$ such that $Y^\top \Delta = 0$. Since $Y$ is not a unique representation, the tangent vector $\Delta$ changes with the choice of $Y$. For example, given a representation $Y_1$ and its tangent vector $\Delta_1$, if a different representation is selected by performing right multiplication, i.e., $Y_2 = Y_1 R$, then the tangent vector must be moved in the same way, that is $\Delta_2 = \Delta_1 R$. The canonical metric, which is invariant under the action of the orthogonal group and scaling [10], is given by

$$g_g(\Delta_1, \Delta_2) = \text{tr}\big((Y^\top Y)^{-1}\Delta_1^\top \Delta_2\big) \tag{4}$$

where $Y^\top \Delta_1 = 0$ and $Y^\top \Delta_2 = 0$. For $\mathcal{G}(1, n)$ with a representation $y$, the metric is given by $g_g(\Delta_1, \Delta_2) = \Delta_1^\top \Delta_2 / y^\top y$. The metric is invariant to the scaling of $y$ as shown below

$$\Delta_1^\top \Delta_2 / y^\top y = (k\Delta_1)^\top (k\Delta_2)/(ky)^\top (ky). \tag{5}$$

Without loss of generality, we can choose a representation with $y^\top y = 1$ to obtain $g_g(\Delta_1, \Delta_2) = \text{tr}(\Delta_1^\top \Delta_2)$, which coincides with the canonical metric for $\mathcal{V}(1, n)$. Hereafter, we will focus on the geometry of $\mathcal{G}(1, n)$ with the metric and representation chosen above, derived from the general formula in [4, 10].

**Gradient of a function**   *The gradient of a function $f(y)$ defined on $\mathcal{G}(1, n)$ is given by*

$$\text{grad} f = g - (y^T g)y \tag{6}$$

*where $g_i = \partial f / \partial y_i$.*

**Exponential map**   *Let $h$ be a tangent vector to $\mathcal{G}(1, n)$ at $y$. The exponential map on $\mathcal{G}(1, n)$ emanating from $y$ with initial velocity $h$ is given by*

$$\exp_y(h) = y \cos|h| + \frac{h}{|h|} \sin|h|. \tag{7}$$

It can be easily shown that $\exp_y(h) = \exp_y((1 + 2\pi/|h|)h)$.

**Parallel translation**   *Let $\Delta$ and $h$ be tangent vectors to $\mathcal{G}(1, n)$ at $y$. The parallel translation of $\Delta$ along the geodesic with the initial velocity $h$ in a unit time is given by*

$$\text{pt}_y(\Delta; h) = \Delta - \big(u(1 - \cos|h|) + y\sin|h|\big)u^\top \Delta, \tag{8}$$

*where $u = h/|h|$.* Note that $|\Delta| = |\text{pt}_y(\Delta; h)|$. If $\Delta = h$, it can be further simplified as

$$\text{pt}_y(h) = h\cos|h| - y|h|\sin|h|. \tag{9}$$

Note that $\text{BN}(z)$ is not invariant to scaling with negative numbers. That is, $\text{BN}(-z) = -\text{BN}(z)$. To be precise, there is an one-to-one mapping between the set of weights on which $\text{BN}(z)$ is invariant and a point on $\mathcal{V}(1, n)$, but not on $\mathcal{G}(1, n)$. However, the proposed method interprets each weight vector as a point on the manifold only when the weight update is performed. As long as the weight vector stays in the domain where $\mathcal{V}(1, n)$ and $\mathcal{G}(1, n)$ have the same invariance property, the weight update remains equivalent. We prefer $\mathcal{G}(1, n)$ since the operators can easily be extended to $\mathcal{G}(p, n)$, opening up further applications.

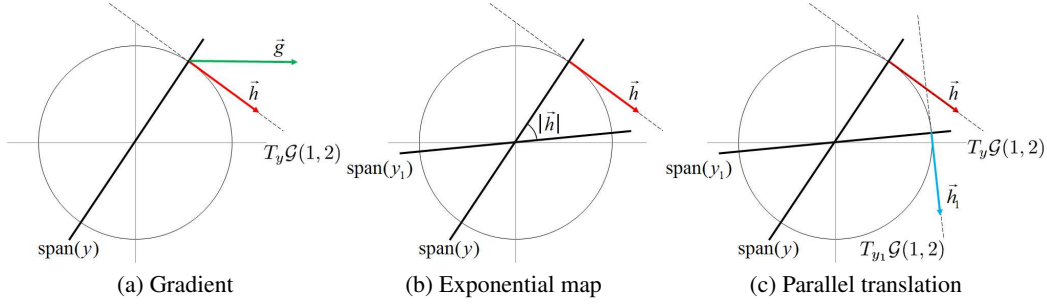

<div align="center">

(a) Gradient       (b) Exponential map       (c) Parallel translation

</div>

Figure 1: An illustration of the operators on the Grassmann manifold $\mathcal{G}(1,2)$. A 2-by-1 matrix $y$ is an orthonormal representation on $\mathcal{G}(1,2)$. (a) A gradient calculated in Euclidean coordinate is projected onto the tangent space $T_y\mathcal{G}(1,2)$. (b) $y_1 = \exp_y(h)$. (c) $h_1 = \mathrm{pt}_y(h)$, $|\vec{h}| = |\vec{h}_1|$.

# 4 Optimization algorithms on $\mathcal{G}(1,n)$

In this section, we derive optimization algorithms on the Grassmann manifold $\mathcal{G}(1,n)$. The algorithms given below are iterative algorithms to solve the following unconstrained optimization:

$$\min_{y \in \mathcal{G}(1,n)} f(y). \tag{10}$$

## 4.1 Stochastic gradient descent with momentum

The application of Algorithm 1 to the Grassmann manifold $\mathcal{G}(1,n)$ is straightforward. We extend this algorithm to the one with momentum to speed up the training [12]. Algorithm 2 presents the pseudo-code of the SGD with momentum on $\mathcal{G}(1,n)$. This algorithm differs from conventional SGD in three ways. First, it projects the gradient onto the tangent space at the point $y$, as shown in Fig. 1 (a). Second, it moves the position by the exponential map in Fig. 1 (b). Third, it moves the momentum by the parallel translation of the Grassmann manifold in Fig. 1 (c). Note that if the weight is initialized with a unit vector, it remains a unit vector after the update.

Algorithm 2 has an advantage over conventional SGD in that the amount of movement is intuitive, i.e., it can be measured by the angle between the original point and the new point. As it returns to the original point after moving by $2\pi$ (radian), it is natural to restrict the maximum movement induced by a gradient to $2\pi$. For first order methods like gradient descent, it would be beneficial to restrict the maximum movement even more so that it stays in the range where linear approximation is valid. Let $h$ be the gradient calculated at $t = 0$. The amount of the first step by the gradient of $h$ is $\delta_0 = \eta \cdot |h|$ and the contributions to later steps are recursively calculated by $\delta_t = \gamma \cdot \delta_{t-1}$. The overall contribution of $h$ is $\sum_{t=0}^{\infty} \delta_t = \eta \cdot |h|/(1 - \gamma)$. In practice, we found it beneficial to restrict this amount to less than $0.2$ (rad) $\cong 11.46°$ by clipping the norm of $h$ at $\nu$. For example, with initial learning rate $\eta = 0.2$, setting $\gamma = 0.9$ and $\nu = 0.1$ guarantees this condition.

---

**Algorithm 2** Stochastic gradient descent with momentum on $\mathcal{G}(1,n)$

---

**Require:** learning rate $\eta$, momentum coefficient $\gamma$, norm_threshold $\nu$
Initialize $y_0 \in \mathbb{R}^{n \times 1}$ with a random unit vector
Initialize $\tau_0 \in \mathbb{R}^{n \times 1}$ with a zero vector
for $t = 1, \cdots, T$
     $g \leftarrow \partial f(y_{t-1})/\partial y$      Run a backward pass to obtain $g$
     $h \leftarrow g - (y_{t-1}^{\top}g)y_{t-1}$      Project $g$ onto the tangent space at $y_{t-1}$
     $\hat{h} \leftarrow \mathrm{norm\_clip}(h, \nu)^{\dagger}$      Clip the norm of the gradient at $\nu$
     $d \leftarrow \gamma\tau_{t-1} - \eta\hat{h}$      Update delta with momentum
     $y_t \leftarrow \exp_{y_{t-1}}(d)$      Move to the new position by the exponential map in Eq. (7)
     $\tau_t \leftarrow \mathrm{pt}_{y_{t-1}}(d)$      Move the momentum by the parallel translation in Eq. (9)
     Note that $h, \hat{h}, d \perp y_{t-1}$ and $\tau_t \perp y_t$ where $h, \hat{h}, d, y_{t-1}, y_t \in \mathbb{R}^{n \times 1}$

---

$^{\dagger}\mathrm{norm\_clip}(h, \nu) = \nu \cdot h/|h|$ if $|h| > \nu$, else $h$

## 4.2 Adam

Adam [9] is a recently developed first-order optimization algorithm based on adaptive estimates of lower-order moments that has been successfully applied to training deep neural networks. In this section, we derive Adam on the Grassmann manifold $\mathcal{G}(1, n)$. Adam computes the individual adaptive learning rate for each parameter. In contrast, we assign one adaptive learning rate to each weight vector that corresponds to a point on the manifold. In this way, the direction of the gradient is not corrupted, and the size of the step is adaptively controlled. The pseudo-code of Adam on $\mathcal{G}(1, n)$ is presented in Algorithm 3.

It was shown in [9] that the effective step size of Adam ($|d|$ in Algorithm 3) has two upper bounds. The first occurs in the most severe case of sparsity, and the upper bound is given as $\eta \frac{1-\beta_1}{\sqrt{1-\beta_2}}$ since the previous momentum terms are negligible. The second case occurs if the gradient remains stationary across time steps, and the upper bound is given as $\eta$. For the common selection of hyperparameters $\beta_1 = 0.9$, $\beta_2 = 0.99$, two upper bounds coincide. In our experiments, $\eta$ was chosen to be 0.05 and the upper bound was $|d| \leq 0.05$ (rad).

---

**Algorithm 3** Adam on $\mathcal{G}(1, n)$

---

**Require:** learning rate $\eta$, momentum coefficients $\beta_1, \beta_2$, norm_threshold $\nu$, scalar $\epsilon = 10^{-8}$
Initialize $y_0 \in \mathbb{R}^{n \times 1}$ with a random unit vector
Initialize $\tau_0 \in \mathbb{R}^{n \times 1}$ with a zero vector
Initialize a scalar $v_0 = 0$
for $t = 1, \cdots, T$

$\quad \eta_t \leftarrow \eta \sqrt{1 - \beta_2^t}/(1 - \beta_1^t)$ $\qquad$ Calculate the bias correction factor
$\quad g \leftarrow \partial f(y_{t-1})/\partial y$ $\qquad$ Run a backward pass to obtain $g$
$\quad h \leftarrow g - (y_{t-1}^\top g)y_{t-1}$ $\qquad$ Project $g$ onto the tangent space at $y_{t-1}$
$\quad \hat{h} \leftarrow \text{norm\_clip}(h, \nu)$ $\qquad$ Clip the norm of the gradient at $\nu$
$\quad m_t \leftarrow \beta_1 \cdot \tau_{t-1} + (1 - \beta_1) \cdot \hat{h}$
$\quad v_t \leftarrow \beta_2 \cdot v_{t-1} + (1 - \beta_2) \cdot \hat{h}^\top \hat{h}$ $\qquad$ ($v_t$ is a scalar)
$\quad d \leftarrow -\eta_t \cdot m_t/\sqrt{v_t + \epsilon}$ $\qquad$ Calculate delta
$\quad y_t \leftarrow \exp_{y_{t-1}}(d)$ $\qquad$ Move to the new point by exponential map in Eq. (7)
$\quad \tau_t \leftarrow \text{pt}_{y_{t-1}}(m_t; d)$ $\qquad$ Move the momentum by parallel translation in Eq. (8)
$\quad$ Note that $h, \hat{h}, m_t, d \perp y_{t-1}$ and $\tau_t \perp y_t$ where $h, \hat{h}, m_t, d, \tau_t, y_{t-1}, y_t \in \mathbb{R}^{n \times 1}$

---

## 5  Batch normalization on the product manifold of $\mathcal{G}(1, \cdot)$

In Sec. 3, we have shown that a weight vector used to compute the pre-activation that serves as an input to the BN transform can be naturally interpreted as a point on $\mathcal{G}(1, n)$. For deep networks with multiple layers and multiple units per layer, there can be multiple weight vectors that the BN transform is applied to. In this case, the training of neural networks is converted into an optimization problem with respect to a set of points on Grassmann manifolds and the remaining set of parameters. It is formalized as

$$\min_{\mathcal{X} \in \mathcal{M}} \mathcal{L}(\mathcal{X}) \quad \text{where} \quad \mathcal{M} = \mathcal{G}(1, n_1) \times \cdots \times \mathcal{G}(1, n_m) \times \mathbb{R}^l \tag{11}$$

where $n_1 \ldots n_m$ are the dimensions of weight vectors, $m$ is the number of the weight vectors on $\mathcal{G}(1, \cdot)$ which will be optimized using Algorithm 2 or 3, and $l$ is the number of remaining parameters which include biases, learnable scaling and offset parameters in BN layers, and other weight matrices.

Algorithm 4 presents the whole process of training deep neural networks. The forward pass and backward pass remain unchanged. The only change made is updating the weights by Algorithm 2 or Algorithm 3. Note that we apply the proposed algorithm only when the input layer to BN is under-complete, that is, the number of output units is smaller than the number of input units, because the regularization algorithm we will derive in Sec. 5.1 is only valid in this case. There should be ways to expand the regularization to over-complete layers. However, we do not elaborate on this topic since 1) the ratio of over-complete layers is very low (under 0.07% for wide resnets and under 5.5% for VGG networks) and 2) we believe that over-complete layers are suboptimal in neural networks, which should be avoided by proper selection of network architectures.

---
**Algorithm 4** Batch normalization on product manifolds of $\mathcal{G}(1, \cdot)$
---
Define the neural network model with BN layers
$m \leftarrow 0$
for $W$ = {weight matrices in the network such that $W^\top x$ is an input to a BN layer}
   Let $W$ be an $n \times p$ matrix
   if $n > p$
     for $i = 1, \cdots, p$
       $m \leftarrow m + 1$
       Assign a column vector $w_i$ in $W$ to $y_m \in \mathcal{G}(1, n)$
Assign remaining parameters to $v \in \mathbb{R}^l$

---
$\min \mathcal{L}(y_1, \cdots, y_m, v)^\dagger$   w.r.t   $y_i \in \mathcal{G}(1, n_i)$ for $i = 1, \cdots, m$ and $v \in \mathbb{R}^l$
for $t = 1, \cdots, T$
   Run a forward pass to calculate $\mathcal{L}$
   Run a backward pass to obtain $\frac{\partial \mathcal{L}}{\partial y_i}$ for $i = 1, \cdots, m$ and $\frac{\partial \mathcal{L}}{\partial v}$
   for $i = 1, \cdots, m$
     Update the point $y_i$ by Algorithm 2 or Algorithm 3
   Update $v$ by conventional optimization algorithms (such as SGD)
---
$\dagger$ For orthogonality regularization as in Sec. 5.1, $\mathcal{L}$ is replaced with $\mathcal{L} + \sum_W L^O(\alpha, W)$

## 5.1 Regularization using variational inference

In conventional neural networks, $L_2$ regularization is normally adopted to regularize the networks. However, it does not work on Grassmann manifolds because the gradient vector of the $L_2$ regularization is perpendicular to the tangent space of the Grassmann manifold. In [13], the $L_2$ regularization was derived based on the Gaussian prior and delta posterior in the framework of variational inference. We extend this theory to Grassmann manifolds in order to derive a proper regularization method in this space.

Consider the complexity loss, which accounts for the cost of describing the network weights. It is given by the Kullback-Leibler divergence between the posterior distribution $Q(w|\beta)$ and the prior distribution $P(w|\alpha)$ [13]:

$$L^C(\alpha, \beta) = D_{KL}(Q(w|\beta) \parallel P(w|\alpha)). \tag{12}$$

Factor analysis (FA) [14] establishes the link between the Grassmann manifold and the space of probabilistic distributions [15]. The factor analyzer is given by

$$p(x) = \mathcal{N}(u, C), \quad C = ZZ^\top + \sigma^2 I \tag{13}$$

where $Z$ is a full-rank $n$-by-$p$ matrix $(n > p)$ and $\mathcal{N}$ denotes a normal distribution. Under the condition that $u = 0$ and $\sigma \to 0$, the samples from the analyzer lie in the linear subspace span$(Z)$. In this way, a linear subspace can be considered as an FA distribution.

Suppose that $n$-dimensional $p$ weight vectors $y_1, \cdots, y_p$ for $n > p$ are in the same layer, which are assumed as $p$ points on $\mathcal{G}(1, n)$. Let $y_i$ be a representation of a point such that $y_i^\top y_i = 1$. With the choice of delta posterior and $\beta = [y_1, \cdots, y_p]$, the corresponding FA distribution can be given by $q(x|Y) = \mathcal{N}(0, YY^\top + \sigma^2 I)$, where $Y = [y_1, \cdots, y_p]$ with the subspace condition $\sigma \to 0$. The FA distribution for the prior is set to $p(x|\alpha) = \mathcal{N}(0, \alpha I)$ that depends on the hyperparameter $\alpha$. Substituting the FA distribution of the prior and posterior into Eq. (12) gives the complexity loss

$$L^C(\alpha, Y) = D_{KL}\big(q(x|Y) \parallel p(x|\alpha)\big). \tag{14}$$

Eq. (14) is minimized when the column vectors of $Y$ are orthogonal to each other (refer to Appendix A for details). That is, minimizing $L^C(\alpha, Y)$ will maximally scatter the points away from each other on $\mathcal{G}(1, n)$. However, it is difficult to estimate its gradient. Alternatively, we minimize

$$L^O(\alpha, Y) = \frac{\alpha}{2} \parallel Y^\top Y - I \parallel_F^2 \tag{15}$$

where $\parallel \cdot \parallel_F$ is the Frobenius norm. It has the same minimum as the original complexity loss and the negative of its gradient is a descent direction of the original loss (refer to Appendix B).

# 6 Experiments

We evaluated the proposed learning algorithm for image classification tasks using three benchmark datasets: CIFAR-10 [16], CIFAR-100 [16], and SVHN (Street View House Number) [17]. We used the VGG network [18] and wide residual network [2, 19, 20] for experiments. The VGG network is a widely used baseline for image classification tasks, while the wide residual network [2] has shown state-of-the-art performance on the benchmark datasets. We followed the experimental setups described in [2] so that the performance of algorithms can be directly compared. Source code is publicly available at https://github.com/MinhyungCho/riemannian-batch-normalization.

CIFAR-10 is a database of 60,000 color images in 10 classes, which consists of 50,000 training images and 10,000 test images. CIFAR-100 is similar to CIFAR-10, except that it has 100 classes and contains fewer images per class. For preprocessing, we normalized the data using the mean and variance calculated from the training set. During training, the images were randomly flipped horizontally, padded by four pixels on each side with the reflection, and a 32×32 crop was randomly sampled. SVHN [17] is a digit classification benchmark dataset that contains 73,257 images in the training set, 26,032 images in the test set, and 531,131 images in the extra set. We merged the extra set and the training set in our experiment, following the step in [2]. The only preprocessing done was to divide the intensity by 255.

Detailed architectures for various VGG networks are described in [18]. We used 512 neurons in fully connected layers rather than 4096 neurons, and the BN layer was placed before every ReLU activation layer. The learnable scaling parameter in the BN layer was set to one because it does not reduce the expressive power of the ReLU layer [21]. For SVHN experiments using VGG networks, the dropout was applied after the pooling layer with dropout rate 0.4. For wide residual networks, we adopted exactly the same model architectures in [2], including the BN and dropout layers. In all cases, the biases were removed except the final layer.

For the baseline, the networks were trained by SGD with Nesterov momentum [22]. The weight decay was set to 0.0005, momentum to 0.9, and minibatch size to 128. For CIFAR experiments, the initial learning rate was set to 0.1 and multiplied by 0.2 at 60, 120, and 160 epochs. It was trained for a total of 200 epochs. For SVHN, the initial learning rate was set to 0.01 and multiplied by 0.1 at 60 and 120 epochs. It was trained for a total of 160 epochs.

For the proposed method, we used different learning rates for the weights in Euclidean space and on Grassmann manifolds. Let us denote the learning rates for Euclidean space and Grassmann manifolds as $\eta_e$ and $\eta_g$, respectively. The selected initial learning rates were $\eta_e = 0.01, \eta_g = 0.2$ for Algorithm 2 and $\eta_e = 0.01, \eta_g = 0.05$ for Algorithm 3. The same initial learning rates were used for all CIFAR experiments. For SVHN, they were scaled by 1/10, following the same ratio as the baseline [2]. The training algorithm for Euclidean parameters was identical to the one used in the baseline with one exception. We did not apply weight decay to scaling and offset parameters of BN, whereas the baseline did as in [2]. To clarify, applying weight decay to mean and variance parameters of BN was essential for reproducing the performance of baseline. The learning rate schedule was also identical to the baseline, both for $\eta_e$ and $\eta_g$. The threshold for clipping the gradient $\nu$ was set to 0.1. The regularization strength $\alpha$ in Eq. (15) was set to 0.1, which gradually achieved near zero $L^O$ during the course of the training, as shown in Fig. 2.

Figure 2: Changes in $L^O$ in Eq. (15) during training for various $\alpha$ values (y-axis on the left) The red dotted line denotes the learning rate ($\eta_g$, y-axis on the right). VGG-11 was trained by SGD-G on CIFAR-10.

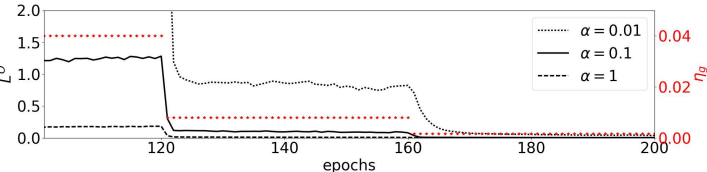

## 6.1 Results

Tables 1 and 2 compare the performance of the baseline SGD and two proposed algorithms described in Sec. 4 and 5, on CIFAR-10, CIFAR-100, and SVHN datasets. All the numbers reported are the median of five independent runs. In most cases, the networks trained using the proposed algorithms

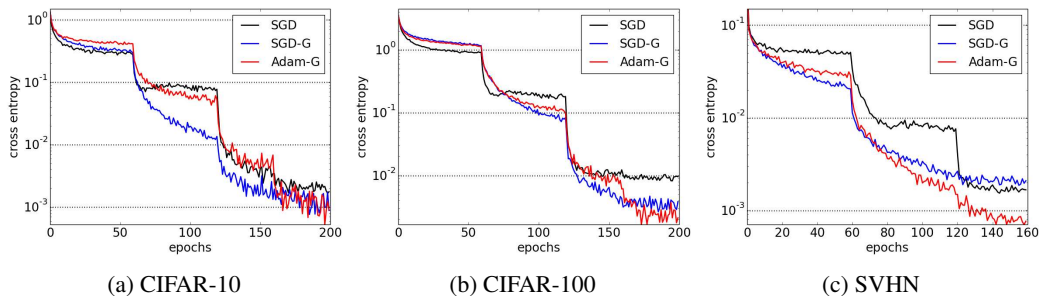

|         | (a) CIFAR-10 | (b) CIFAR-100 | (c) SVHN |

Figure 3: Training curves of the baseline and proposed optimization methods. (a) WRN-28-10 on CIFAR-10. (b) WRN-28-10 on CIFAR-100. (c) WRN-22-8 on SVHN.

outperformed the baseline across various datasets and network configurations, especially for the ones with more parameters. The best performance was 3.72% (SGD and SGD-G) and 17.85% (ADAM-G) on CIFAR-10 and CIFAR-100, respectively, with WRN-40-10; and 1.55% (ADAM-G) on SVHN with WRN-22-8.

Training curves of the baseline and proposed methods are presented in Figure 3. The training curves for SGD suffer from instability or experience a plateau after each learning rate drop, compared to the proposed methods. We believe that this comes from the inverse proportionality of the gradient to the norm of BN weight parameters (as in Eq. (2)). During the training process, this norm is affected by weight decay, hence the magnitude of the gradient. It is effectively equivalent to disturbing the learning rate by weight decay. The authors of wide resnet also observed that applying weight decay caused this phenomena, but weight decay was indispensable for achieving the reported performance [2]. Proposed methods resolve this issue in a principled way.

Table 3 summarizes the performance of recently published algorithms on the same datasets. We present the best performance of five independent runs in this table.

Table 1: Classification error rate of various networks on CIFAR-10 and CIFAR-100 (median of five runs). VGG-$l$ denotes a VGG network with $l$ layers. WRN-$d$-$k$ denotes a wide residual network that has $d$ convolutional layers and a widening factor $k$. SGD-G and Adam-G denote Algorithm 2 and Algorithm 3, respectively. The results in parenthesis show those reported in [2].

| Dataset | CIFAR-10 | | | CIFAR-100 | | |
|---------|----------|--------|--------|-----------|--------|--------|
| Model | SGD | SGD-G | Adam-G | SGD | SGD-G | Adam-G |
| VGG-11 | 7.43 | 7.14 | 7.59 | 29.25 | 28.02 | 28.05 |
| VGG-13 | 5.88 | 5.87 | 6.05 | 26.17 | 25.29 | 24.89 |
| VGG-16 | 6.32 | 5.88 | 5.98 | 26.84 | 25.64 | 25.29 |
| VGG-19 | 6.49 | 5.92 | 6.02 | 27.62 | 25.79 | 25.59 |
| WRN-52-1 | 6.23 (6.28) | 6.56 | 6.58 | 27.44 (29.78) | 28.13 | 28.16 |
| WRN-16-4 | 4.96 (5.24) | 5.35 | 5.28 | 23.41 (23.91) | 24.51 | 24.24 |
| WRN-28-10 | 3.89 (3.89) | 3.85 | 3.78 | 18.66 (18.85) | 18.19 | 18.30 |
| WRN-40-10[†] | **3.72** (3.8) | **3.72** | 3.80 | 18.39 (18.3) | 18.04 | **17.85** |

[†]This model was trained on two GPUs. The gradients were summed from two minibatches of size 64, and BN statistics were calculated from each minibatch.

## 7  Conclusion and discussion

We presented new optimization algorithms for scale-invariant vectors by representing them on $\mathcal{G}(1, n)$ and following the intrinsic geometry. Specifically, we derived SGD with momentum and Adam algorithms on $\mathcal{G}(1, n)$. An efficient regularization algorithm in this space has also been proposed. Applying them in the context of BN showed consistent performance improvements over the baseline BN algorithm with SGD on CIFAR-10, CIFAR-100, and SVHN datasets.

Table 2: Classification error rate of various networks on SVHN (median of five runs).

| Model | SGD | SGD-G | Adam-G |
|---|---|---|---|
| VGG-11 | 2.11 | 2.10 | 2.14 |
| VGG-13 | 1.78 | 1.74 | 1.72 |
| VGG-16 | 1.85 | 1.76 | 1.76 |
| VGG-19 | 1.94 | 1.81 | 1.77 |
| WRN-52-1 | 1.68 (1.70) | 1.72 | 1.67 |
| WRN-16-4 | 1.64 (1.64) | 1.67 | 1.61 |
| WRN-16-8 | 1.60 (1.54) | 1.69 | 1.68 |
| WRN-22-8 | 1.64 | 1.63 | **1.55** |

Table 3: Performance comparison with previously published results.

| Method | CIFAR-10 | CIFAR-100 | SVHN |
|---|---|---|---|
| NormProp [7] | 7.47 | 29.24 | 1.88 |
| ELU [23] | 6.55 | 24.28 | - |
| Scalable Bayesian optimization [24] | 6.37 | 27.4 | - |
| Generalizing pooling [25] | 6.05 | - | 1.69 |
| Stochastic depth [26] | 4.91 | 24.98 | 1.75 |
| ResNet-1001 [20] | 4.62 | 22.71 | - |
| Wide residual network [2] | 3.8 | 18.3 | 1.54 |
| Proposed (best of five runs) | **3.49**[1] | **17.59**[2] | **1.49**[3] |

[1]WRN-40-10+SGD-G [2]WRN-40-10+Adam-G [3]WRN-22-8+Adam-G

Our work interprets each scale invariant piece of the weight matrix as a separate manifold, whereas natural gradient based algorithms [27, 28, 29] interpret the whole parameter space as a manifold and constrain the shape of the cost function (i.e. to the KL divergence) to obtain a cost efficient metric. There are similar approaches to ours such as Path-SGD [30] and the one based on symmetry-invariant updates [31], but the comparison remains to be done.

Proposed algorithms are computationally as efficient as their non-manifold versions since they do not affect the forward and backward propagation steps, where majority of the computation takes place. The weight update step is 2.5-3.5 times more expensive, but still $O(n)$.

We did not explore the full range of possibilities offered by the proposed algorithm. For example, techniques similar to BN, such as weight normalization [6] and normalization propagation [7], have scale invariant weight vectors and can benefit from the proposed algorithm in the same way. Layer normalization [8] is invariant to weight matrix rescaling, and simple vectorization of the weight matrix enables the application of the proposed algorithm.

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
