[Supplementary Material · riemannian-batch-normalization-supplementary.pdf]

# Riemannian approach to batch normalization: Supplementary material

**Minhyung Cho**    **Jaehyung Lee**
Applied Research Korea, Gracenote Inc.
mhyung.cho@gmail.com    jaehyung.lee@kaist.ac.kr

## A    Minimum of the complexity loss $L^C(\alpha, Y)$

The symmetric KL divergence between two $n$-dimensional normal distributions $\mathcal{N}_0(u_0, C_0)$, $\mathcal{N}_1(u_1, C_1)$ is given by

$$D_{KL}(\mathcal{N}_0 \parallel \mathcal{N}_1) = \frac{1}{2}\mathrm{tr}(C_1^{-1}C_0 + C_0^{-1}C_1) + (u_1 - u_0)^\top (C_0^{-1} + C_1^{-1})(u_1 - u_0) - n. \quad (16)$$

Recall from Eq. (14) in Sec. 5.1, that is,

$$L^C = D_{KL}\big(q(x|Y) \parallel p(x|\alpha)\big) \quad (17)$$

where $q(x|Y) = \mathcal{N}(0, \sigma^2 I + YY^\top)$, $Y \in \mathbb{R}^{n \times p}$, $n > p$, each column of $Y$ is normalized to one, and $p(x|\alpha) = \mathcal{N}(0, \alpha I)$. Substituting $u_0 = 0$, $u_1 = 0$, $C_0 = \sigma^2 I + YY^\top$, and $C_1 = \alpha I$ into Eq. (16) gives

$$L^C = \frac{1}{2}\mathrm{tr}\big(\alpha(\sigma^2 I + YY^\top)^{-1} + \frac{1}{\alpha}(\sigma^2 I + YY^\top)\big) - n. \quad (18)$$

The second term in the right-hand side of Eq. (18) is a constant as shown below:

$$\frac{1}{\alpha}\mathrm{tr}(\sigma^2 I + YY^\top) = \frac{\sigma^2\mathrm{tr}(I) + \mathrm{tr}(Y^\top Y)}{\alpha} = \frac{\sigma^2 n + p}{\alpha}. \quad (19)$$

After removing the constant terms in Eq. (18), we obtain

$$L^C = \frac{\alpha}{2}\mathrm{tr}\big((\sigma^2 I + YY^\top)^{-1}\big). \quad (20)$$

The following propositions are used to prove that $L^C$ is minimized when the column vectors of $Y$ are orthogonal to each other.

Prop. 1   Suppose $A \in \mathbb{R}^{n \times n}$ is an invertible matrix and $y \in \mathbb{R}^{n \times 1}$ is a column vector. If $(A + yy^\top)^{-1}$ is invertible, $(A + yy^\top)^{-1} = A^{-1} - \frac{A^{-1}yy^\top A^{-1}}{1 + y^\top A^{-1}y}$ [1].

Prop. 2   Let $B \in \mathbb{R}^{n \times p}$ be a full rank matrix where $n > p$. The eigenvalues of $\sigma^2 I + BB^\top$ are given by $\{\sigma^2 + \lambda_1, \cdots, \sigma^2 + \lambda_p, \cdots, \sigma^2\}$ where $\lambda_1, \cdots, \lambda_p$ are the eigenvalues of $B^\top B$.

Prop. 3   Let $B \in \mathbb{R}^{n \times p}$ be a full rank matrix where $n > p$ and $\beta \in \mathbb{R}^{n \times p}$ contain eigenvectors of $\sigma^2 I + BB^\top$ corresponding to $p$ largest eigenvalues in its columns, then $\mathrm{span}(\beta) = \mathrm{span}(B)$.

Prop. 4   If two positive definite matrices $P$, $Q$ share the eigenvectors, $\min_y \frac{y^\top P y}{y^\top Q y}$ and $\min_y \frac{y^\top Q^{-1} P y}{y^\top y}$ have the same minimum.

It is straightforward to derive Prop. 2 and 3 (refer to [2] for the general idea). Prop. 4 comes from the solution of generalized eigenvalue problems.

Let $X \in \mathbb{R}^{n \times (p-1)}$ be a matrix obtained by omitting a column vector $y$ from $Y \in \mathbb{R}^{n \times p}$ in Eq. (20). First, we show that the minimum of $L^C$ with respect to $y$ is achieved when $y$ is orthogonal to $\text{span}(X)$.

It can be easily shown that $YY^\top = XX^\top + yy^\top$. Let $Z = \sigma^2 I + XX^\top$. Then $Z + yy^\top = \sigma^2 I + YY^\top$ is invertible. From Prop. 1, we obtain

$$\text{tr}\big((Z + yy^\top)^{-1}\big) = \text{tr}(Z^{-1}) - \frac{\text{tr}(Z^{-1}yy^\top Z^{-1})}{1 + y^\top Z^{-1}y}. \tag{21}$$

The numerator of the rightmost term can be rewritten as $\text{tr}(Z^{-1}yy^\top Z^{-1}) = y^\top Z^{-1}Z^{-1}y$ because the trace is invariant under cyclic permutations. The denominator can be rewritten as $1 + y^\top Z^{-1}y = y^\top(I + Z^{-1})y$ because $y^\top y = 1$. Since $\text{tr}(Z^{-1})$ has no dependency on $y$, minimizing Eq. (21) with respect to $y$ is equivalent to

$$\min_y -\frac{y^\top Z^{-1}Z^{-1}y}{y^\top(I + Z^{-1})y}, \tag{22}$$

which is the generalized Rayleigh quotient. The eigenvectors of $Z^{-1}Z^{-1}$ and $I + Z^{-1}$ are the same. Applying Prop. 4 (note that $y^\top y = 1$) yields

$$\min_y -y^\top(I + Z^{-1})^{-1}Z^{-1}Z^{-1}y. \tag{23}$$

Let the eigenvalues of $X$ be $\{\lambda_1, \cdots, \lambda_{p-1}\}$ where $\lambda_1 > \cdots > \lambda_{p-1}$. From Prop. 2, the eigendecomposition of $Z$ is given by

$$Z = V\Sigma V^\top \tag{24}$$

where $\Sigma = \text{diag}[\sigma^2 + \lambda_1, \cdots, \sigma^2 + \lambda_{p-1}, \sigma^2, \cdots, \sigma^2]$ and $V$ is the corresponding eigenvector matrix. From this, we have $I + Z^{-1} = V(I + \Sigma^{-1})V^\top$, $Z^{-1}Z^{-1} = V\Sigma^{-1}\Sigma^{-1}V^\top$, and $(I + Z^{-1})^{-1}Z^{-1}Z^{-1} = V(I + \Sigma^{-1})^{-1}\Sigma^{-1}\Sigma^{-1}V^\top$. With $y^\top y = 1$ and $\Sigma = \text{diag}[\sigma^2 + \lambda_1, \cdots, \sigma^2 + \lambda_{p-1}, \cdots, \sigma^2]$, Eq. (23) is rewritten as

$$-y^\top(I + Z^{-1})^{-1}Z^{-1}Z^{-1}y = -\sum_{i=1}^{p-1} \frac{(v_i^\top y)^2}{(\sigma^2 + \lambda_i)(\sigma^2 + \lambda_i + 1)} - \sum_{i=p}^{n} \frac{(v_i^\top y)^2}{\sigma^2(\sigma^2 + 1)} \tag{25}$$

where $v_i$ is the eigenvector of $Z$ corresponding to the $i$-th largest eigenvalue. With the subspace condition $\sigma^2 \to 0$, the first term in the right-hand side can be ignored. By dropping the constant factor and applying $\sum_{i=1}^{n}(v_i^\top y)^2 = 1$, the optimization in Eq. (23) reduces to

$$\min_y \sum_{i=1}^{p-1}(v_i^\top y)^2. \tag{26}$$

From Prop. 3, $\text{span}([v_1, \cdots, v_{p-1}]) = \text{span}(X)$. Thus, $L^C$ is minimized when $y$ is orthogonal to $\text{span}(X)$.

It follows that if all the column vectors of $Y$ are orthogonal to each other, $L^C$ is minimized with respect to all the parameters in $Y$. Since it is always possible to find such a set of column vectors if $n > p$, it is guaranteed that the minimum of $L^C$ can be reached in any case.

## B  The direction of the gradients of $L^C(\alpha, Y)$ and $L^O(\alpha, Y)$

In this section, we show that the negative of the gradient of the regularization loss $L^O(\alpha, Y)$ in Eq. (15) is a descent direction of the original objective $L^C(\alpha, Y)$ in Eq. (14). Specifically, the inner product of their gradients is shown to be nonnegative. We start the proof from Eq. (22), which is equivalent to $L^C(\alpha, Y)$ except constant terms. Taking the partial derivative of Eq. (22) with respect to a selected column vector $y$ gives

$$\frac{\partial L^C(\alpha, y)}{\partial y} = \frac{-(y^\top Qy)Py + (y^\top Py)Qy}{(y^\top Qy)^2} \tag{27}$$

where $P = Z^{-1}Z^{-1}$ and $Q = I + Z^{-1}$.

The eigendecompositions of $P$ and $Q$ can be derived from Eq. (24) as follows: $P = V\Sigma_1 V^\top$ and $Q = V\Sigma_2 V^\top$ where

$$\Sigma_1 = \text{diag}\left[\frac{1}{(\sigma^2 + \lambda_1)^2}, \cdots, \frac{1}{(\sigma^2 + \lambda_{p-1})^2}, \cdots, \frac{1}{\sigma^4}\right] \tag{28}$$

and

$$\Sigma_2 = \text{diag}\left[1 + \frac{1}{\sigma^2 + \lambda_1}, \cdots, 1 + \frac{1}{\sigma^2 + \lambda_{p-1}}, \cdots, 1 + \frac{1}{\sigma^2}\right]. \tag{29}$$

It follows that $Py$, $Qy$, $y^\top Py$, and $y^\top Qy$ can be computed as:

$$Py = \sum_{i=1}^{p-1} \frac{v_i^\top y}{(\sigma^2 + \lambda_i)^2} v_i + \frac{1}{\sigma^4} \sum_{i=p}^{n} (v_i^\top y) v_i \tag{30}$$

$$y^\top Py = \sum_{i=1}^{p-1} \frac{(v_i^\top y)^2}{(\sigma^2 + \lambda_i)^2} + \frac{1}{\sigma^4} \sum_{i=p}^{n} (v_i^\top y)^2 = r \tag{31}$$

$$Qy = y + \sum_{i=1}^{p-1} \frac{v_i^\top y}{\sigma^2 + \lambda_i} v_i + \frac{1}{\sigma^2} \sum_{i=p}^{n} (v_i^\top y) v_i \tag{32}$$

$$y^\top Qy = 1 + \sum_{i=1}^{p-1} \frac{(v_i^\top y)^2}{\sigma^2 + \lambda_i} + \frac{1}{\sigma^2} \sum_{i=p}^{n} (v_i^\top y)^2 = w \tag{33}$$

where $v_i$ is the $i$-th column of $V$ in Eq. (24). Note that $r$ and $w$ are greater than zero because $P$ and $Q$ are positive definite.

We are only interested in the direction of the gradient. The denominator of Eq. (27) can be discarded since it is always positive. Substituting the equations above into the numerator of Eq. (27) yields

$$g_1 = ry + \sum_{i=1}^{p-1} \left(-\frac{w}{(\sigma^2 + \lambda_i)^2} + \frac{r}{\sigma^2 + \lambda_i}\right)(v_i^\top y) v_i + \left(-\frac{w}{\sigma^4} + \frac{r}{\sigma^2}\right) \sum_{i=p}^{n} (v_i^\top y) v_i \tag{34}$$

where $r$ and $w$ are given in Eq. (31) and (33).

On the other hand, the regularization loss function in Eq. (15) can be rewritten with respect to the same selected column vector $y$ as follows, given that $YY^\top = XX^\top + yy^\top$ and $y^\top y = 1$:

$$L^O(\alpha, Y) = \frac{\alpha}{2} \| Y^\top Y - I \|_F^2 \tag{35}$$

$$= \frac{\alpha}{2} \left\{ \text{tr}(Y^\top YY^\top Y) - 2\text{tr}(Y^\top Y) + \text{tr}(I) \right\} \tag{36}$$

$$= \frac{\alpha}{2} \left\{ \text{tr}(XX^\top XX^\top) + 2y^\top XX^\top y + y^\top y - 2\text{tr}(XX^\top) - 2y^\top y + \text{tr}(I) \right\} \tag{37}$$

$$= \alpha y^\top XX^\top y + \frac{\alpha}{2} \left\{ \text{tr}(XX^\top XX^\top) - 2\text{tr}(XX^\top) + \text{tr}(I) - 1 \right\}. \tag{38}$$

Taking the derivative with respect to $y$ and ignoring the constants, the direction of the gradient of $L^O(\alpha, Y)$ is given by

$$g_2 = XX^\top y = \sum_{i=1}^{p-1} \lambda_i (v_i^\top y) v_i. \tag{39}$$

Taking the inner product of $g_1$ and $g_2$ yields

$$g_1^\top g_2 = r \sum_{i=1}^{p-1} \lambda_i (v_i^\top y)^2 + \sum_{i=1}^{p-1} \lambda_i \left(-\frac{w}{(\sigma^2 + \lambda_i)^2} + \frac{r}{\sigma^2 + \lambda_i}\right)(v_i^\top y)^2 v_i^\top v_i. \tag{40}$$

The first term is nonnegative since $\lambda_i$ and $r$ are nonnegative. To see the second term is nonnegative, we need to show $\frac{r}{\sigma^2 + \lambda_i} > \frac{w}{(\sigma^2 + \lambda_i)^2}$. Since $Y$ is a full-rank $n$-by-$p$ matrix, there exists an $i \in \{p \ldots n\}$ such that $v_i^\top y$ is nonzero. It follows that $\sum_{i=p}^{n} (v_i^\top y)^2 > 0$. Under the subspace condition $\sigma \to 0$, it is easily shown that $r \gg w$. Since $\lambda_i$ are finite numbers, we obtain $\frac{r}{\sigma^2 + \lambda_i} \gg \frac{w}{(\sigma^2 + \lambda_i)^2}$.

From above, we have shown that $g_1^\top g_2 \geq 0$. The equality holds when $\sum_{i=1}^{p-1} (v_i^\top y)^2 = 0$ (that is, the column vector $y$ is orthogonal to all the other column vectors in $Y$). Therefore, the negative of the gradient of the regularization loss $L^O(\alpha, Y)$ in Eq. (15) is a descent direction of the original objective $L^C(\alpha, Y)$ in Eq. (14).