[Reviews · NeurIPS 2017]

Reviewer 1



Paper Summary Starting from the observation that batch-normalization induces a particular form of scale invariance on the weight matrix, the authors propose instead to directly learn the weights on the unit-sphere. This is motivated from information geometry as an example of optimization on a Riemannian manifold, in particular the Stiefel manifold V(1,n) which contains unit-length vectors. As the descent direction on the unit sphere is well known (eq 7), the main contribution of the paper is in extending popular optimization algorithms (SGD+momentum and Adam) to constrained optimization on the unit-sphere. Furthermore, the authors propose orthogonality as a (principled) replacement for L2 regularization, which is no longer meaningful with norm constraints. The method is shown to be effective across two families of models (VGG, wide resnet) on CIFAR-10, CIFAR-100 and SVHN. Review I like this paper, as it proposes a principled alternative to batch normalization, which is also computationally lightweight, compared to other manifold optimization algorithms (e.g. natural gradient). That being said a few flaws prevent me from giving a clear signal for acceptance. (1) Clarity. I believe the paper would greatly benefit from increased simplicity and clarity in the exposition of the idea. First and foremost, concepts of element-wise scale invariance and optimization under unit-norm constraints are much simpler to grasp (especially to people not familiar with information geometry) than Riemannian manifolds, exponential maps and parallel transport. I recommend the authors preface exposition of their methods with such an intuitive explanation. Second, I also found there was a general lack of clarity about what type of invariance is afforded by BN, and which direction were the weights norms constrained in (column vs row). On line 75, the authors write that "w^T x is invariant to linear scaling", when really this refers to element-wise scaling of the columns of the matrix W \in mathbb{R}^{n x p}, where $p$ is the number of hidden units. In Algorithm 4, there is also a discrepancy between the product "W x" (implies W is p x n) and the "column vectors w_i" being unit-norm. (2) Experimental results. As this is inherently an optimization algorithm, I would expect there to be training curves. Does the algorithm simply converge faster, does it yield a better minimum, or both ? Also, I would like to see a controlled experiment showing the impact of the orthogonality regularizer: what is the test error with and without ? (3) Baselines and related work. This work ought to be discussed in the more general context of Riemannian optimization for neural networks, e.g. natural gradient descent algorithms (TONGA, K-FAC, HF, natural nets, etc.). Path-SGD also seems closely related and should be featured as a baseline. Finally, I am somewhat alarmed and puzzled that the authors chose to revert to standard SGD for over-complete layers, and only use their method in the under-complete case. Could the authors comment on how many parameters are in this "under-complete" regime vs total number of parameters ? Are there no appropriate relaxations of orthogonality ? I would have much rather seen the training curves using their algorithm for *all* parameters (never mind test error for now) then this strange hybrid method.

Reviewer 2



The paper proposes a Riemannian approach to batch normalization and suggests to modify SGD with momentum and Adam to benefit from invariances of the proposed approach. The paper is easy to follow but the validation of the proposed approach looks questionable to me. First, the results reported in Table 1 and Table 2 can be a result of better hyperparameter selection. For instance, the results obtained for WRN-28-10 on CIFAR-10 and CIFAR-100 do not differ sufficiently from the ones reported in the original WRNs paper. The authors do not reproduce the results of the original paper but obtain worse results in most cases. This might suggest that the experimental settings are not the same. Indeed, the authors report that they use batchsize 100 which alone might have some contribution to the observed differences. The authors do not report how many runs they performed so I assume that they perform just one run. In contrast to the default SGD with momentum, the proposed methods include a gradient clipping procedure which is known to have certain regularization effects. It is unclear whether the reported results can be explained by the gradient clipping alone. The way how the results are reported looks very suboptimal to me. First, it is of interest to know whether the proposed approaches faster optimizes the training loss due to the proposed batch normalization. Alternatively, it might be that they are not faster in terms of optimization but the networks generalize better. Then, it would be of interest to clarify which process contributes most to better generalization if any. Unfortunately, both the main paper and supplementary material does not contain any details about it. Without having it, it might be assumed that the proposed methods include known useful tricks like weight clipping + involve attached hyperparameters (line 133) whose careful manual selection allowed to get slightly better results in a couple of runs. Please try to refute the latter hypothesis. "Note that we apply the proposed algorithm only when the input layer to BN is under-complete, that is, the number of output units is smaller than the number of input units, because the regularization algorithm we will derive in Sec. 5.1 is only valid in this case." Could you please clarify how often it is the case in practice? The paper (or supp. material) is missing details about the time cost of the proposed methods w.r.t. the default SGD with momentum. I guess that some non-negligible overhead might be expected. Finally, without further clarifications of the experimental part it is hard to asses the impact of the proposed method. Just showing (without a closer look) slightly better results on the CIFAR datasets is of limited importance because better results are already available, e.g., below 3\% for CIFAR-10 with the "Shake-Shake regulation" by Gasteldi, ICLR-2017.

Reviewer 3



The paper presents a novel training algorithm for feedforward network to be used in presence of batch normalization. The purpose of the algorithm is to takle the presence of invariance in the network, by using a geometric approach based on manifold optimization, in particular using Grassmann manifold optimization methods. The paper is well written and sound. The experimental part is convincing and experiments show good performances. I recommend acceptance for the conference. In the following some remarks and comments. In Algorithm 1 you use g for the Riemannian gradient. In line 100, you use g for the vector of partial derivatives. As in Absil et al., I personally prefer "grad f" to represent the Riemannian gradient. For eqs 6-8 I would add an explicit reference, as done for eq 3. In the definition of the parallel transport (line 104), it is not explicit in my opinion the dependence of the operator on the point along the geodesic where you transport the vector: pt_y(\Delta, h), should also depend on t, which parametrize the geodetic gamma(t). The paragraph lines 107-112 makes an important point about invariance, which is addressed by the comment "our purpose is to locally compute the gradient direction using the operators defined above". This is not clear to me, how does it solve the problem of the non invariance with negative numbers. Can you please better explain this part? In line 210 you mention the use of dropout in combination with batch normalization. In the original batch normalization paper, the authors claim dropout is not necessarily in presence of batch normalization. Did you experiment this in your experiments? Can you comment on that? When you pad the images in CIFAR-10, how do you fill the new pixel? Are they left black? As to experimental part, can you please mention how many runs (executions) did you do? Can you mention variance? Another important point is related to time complexity of the algorithm and the impact of the Riemannian approach compared to an Euclidean one, in terms of execution time for the experiments in Table 1. Section 5.1 seems somehow a little bit disconnected from the rest of the paper. Let me better argument. The authors use FA to link the Grassmann manifold to the space of probabilistic function, and this give them a nice way to measure the KL divergence between the posterior and the prior. However in practice they say they use a different function, which appears in 15. My question is if the analysis based on FA can be useful in practice, or if gives just a function intersting from a theoretical perspective and but usable in practice, and if 15 could be instead introduced directly without referring to FA (and to the Grassmann manifold). Moreover, how do you use L^C in training? Does it play a role of regularizing term? I would add a reference to that in the Algorithm 4 you have in the paper. Finally, in Figure 2, how do you change the learning rate in the experiments? can you plot it over the same plot?